# Saikosaponin-b2 Inhibits Primary Liver Cancer by Regulating the STK4/IRAK1/NF-κB Pathway

**DOI:** 10.3390/biomedicines11102859

**Published:** 2023-10-22

**Authors:** Chanhao Lei, Zihan Gao, Xingzhi Lv, Yanxue Zhu, Ruifang Li, Sanqiang Li

**Affiliations:** Department of Pharmacology, Basic Medical College, Henan University of Science and Technology, KaiYuan Avenue 263, Luoyang 471023, China; leila0615@163.com (C.L.); 13643796649@163.com (Z.G.); xingzhilyu@163.com (X.L.); 15515343013@163.com (Y.Z.)

**Keywords:** primary liver cancer, saikosaponin-b2, serine/threonine protein kinase 4, interleukininterleukin-1 receptor-associated kinase 1, nuclear factor-kappa B

## Abstract

The development of primary liver cancer (PLC) is associated with chronic liver inflammation and the loss of associated tumor suppressor genes, which characterizes inflammation-related tumors. In this study, we aimed to explore the effect of saikosaponin-b2 (SS-b2) on the development of PLC and its effect of the STK4 expression and IRAK1/NF-κB signaling axis. In vitro and in vivo experiments showed that SS-b2 exerted potent anti-inflammatory and antitumor effects. A PLC model was induced in vivo by treating male BALB/c mice with diethylnitrosamine, while an inflammatory model was induced in vitro by exposing RAW 264.7 macrophages to lipopolysaccharides (LPS). After treating cancer mice with SS-b2, the serum levels of alpha-fetoprotein, aspartate aminotransferase, alanine aminotransferase, and lactate dehydrogenase significantly reduced. Ki67 expression also decreased. The carcinomatous lesions of the liver were attenuated. Similar results were observed in liver tissue and RAW 264.7 macrophages, where SS-b2 significantly elevated serine/threonine protein kinase 4 (STK4) expression and decreased the expression of interleukin-1 receptor–associated kinase 1 (IRAK1), nuclear factor-kappaB (NF-κB), and downstream inflammatory cytokines, thus exerting anti-cancer and anti-inflammatory effects. Moreover, we employed siRNA to silence the STK4 expression in HepG2 to investigate the anti-tumor effect of SS-b2 in vitro. The STK4 knockdown would upregulate IRAK1 and thus the activation of NF-κB activity revealed by the increase in the levels of proinflammatory cytokines, consequently impairing SS-b2-induced inhibition of liver cancer development. Consequently, SS-b2 effectively inhibited PLC by upregulating STK4 to suppress the IRAK1/NF-κB signaling axis and is a promising agent for treating this disease.

## 1. Introduction

Primary liver cancer (PLC) stands as the second leading cause of cancer-related deaths in China [1]. PLC encompasses hepatocellular carcinoma (HCC) and intrahepatic cholangiocarcinoma, characterized by high incidence and mortality rates. HCC, constituting 80% to 85% of PLC cases, primarily arises due to chronic viral hepatitis B and C, alcoholic and nonalcoholic steatohepatitis, and exposure to aflatoxins. PLC carries an unfavorable prognosis post-surgery, marked by a high propensity for metastasis and recurrence [2,3]. Recent investigations have unveiled the profound impact of inflammation on tumor development, with chronic inflammation fostering tumor progression and resistance to treatment [4]. Consequently, inflammatory conditions leading to chronic liver disease elevate the risk of liver cancer. Prolonged inflammatory responses can incite gene mutations, acting as precursors to PLC, while also disrupting immune system function, thereby altering the liver microenvironment and promoting tumor cell proliferation. Recent studies have even identified certain tumor suppressor genes expressed in immune cells, prompting the intriguing question: Can genes possessing anti-inflammatory and anticancer properties be pinpointed as novel drug targets? Targeting these signaling molecules in both immune and tumor cells may wield a profound influence on cancer development, especially in inflammation-related cancers.

Interleukin-1 receptor-associated kinase 1 (IRAK1), a member of the IRAK kinase family, plays a crucial role in the innate immune system through Toll-like receptors (TLRs) and interleukin-1 receptor (IL-1R)-mediated signaling pathways [5]. Research has elucidated that Toll-like receptors (TLRs), upon binding with their specific ligands, can recruit and activate IRAK1 as a signaling protein. Subsequently, the binding site of I-κB and NF-κB is activated, facilitating the translocation of NF-κB into the nucleus and initiating the expression of downstream inflammatory factors, including interleukins and tumor necrosis factor-alpha (TNF-α), as well as the transcription of genes associated with cell apoptosis [6,7]. Serine/threonine protein kinase 4 (*STK4*), a pivotal regulator of the Hippo–YAP pathway, plays an important role in suppressing tumor cell growth [8]. Deficiency in *STK4* leads to an increase in liver size and promotes hepatocellular carcinoma (HCC) [9]. Studies have demonstrated that the Hippo signaling pathway functions as a tumor-suppressive mechanism in the mammalian liver. Kim et al. found that enhancing Hippo signaling in hepatocytes and macrophages may decrease hepatocyte proliferation, inflammation, and macrophage infiltration, resulting in a reduction in tumor formation [10].

Moreover, earlier research has indicated that STK4 can bind to IRAK1, facilitating its phosphorylation and degradation, thereby inhibiting the activation of NF-κB signaling and the production of proinflammatory cytokines [11]. Our prior investigation also noted elevated IRAK1 expression in clinical HCC patients, which correlated with an unfavorable prognosis. IRAK1 holds potential as a prognosis indicator and therapeutic target for HCC [12]. Given the pivotal roles of STK4 and IRAK1 in the production of proinflammatory cytokines and cell proliferation in liver tumor cells, targeting STK4 and IRAK1 may present a novel strategy for inhibiting the development of inflammation-related liver cancer.

*Bupleurum*, a traditional Chinese medicinal herb, contains saikosaponin (SS) as its primary active ingredient. Saikosaponin (SS) has a wide range of effects, including anti-inflammatory and hepatoprotective properties [13], as well as antitumor activity [14], antidepressant effects [15], anti-infective properties, and hypolipidemic effects [16]. Recent studies on SSs have primarily focused on SS-a and SS-d, both of which exhibit anticancer effects through various mechanisms [17,18]. These compounds can inhibit NF-κB and its downstream inflammatory factors, thereby suppressing the proliferation, angiogenesis, and invasion of liver cancer cells while promoting apoptosis [19,20,21]. Our previous study indicated that SS-b2 possesses hepatoprotective, anti-inflammatory, and antitumor effects. However, the impact of SS-b2 on PLC remains unexplored. In this experiment, we investigated the anti-inflammatory and anticancer effects of SS-b2 on PLC, along with its regulation of the STK4 expression and IRAK1/NF-κB signaling axis. These findings provide novel insights for potential clinical treatments for PLC.

## 2. Materials and Methods

### 2.1. Drugs and Reagents

SS-b2 (purity ≥ 98%, cat. no. MUST-18032104) was purchased from Chengdu MUST Biotechnology Co., Ltd. (Chengdu, China) and dissolved in dimethyl sulfoxide (DMSO) (the final concentration of DMSO in all experiments did not exceed 0.05%). Doxorubicin (DOX, purity > 99%, cat. no. H44024359) was purchased from Shenzhen Main Luck Pharmaceuticals Inc., Shenzhen, China. Diethylnitrosamine (DEN, purity > 99%, cat. no. N109571) was purchased from Shanghai Aladdin Bio-Chem Technology Co., Ltd., Shanghai, China. Lipopolysaccharides (LPS, purity ≥ 98%, cat. no. L8880) were purchased from Beijing Solarbio Science & Technology Co., Ltd., Beijing, China. Dexamethasone (DEX, cat. no. H41020056) was purchased from Zhengzhou Zhuofeng Pharmaceutical Co., Ltd., Zhengzhou, China. The alanine aminotransferase (ALT, cat. no. 20220425), aspartate aminotransferase (AST, cat. no. 20220424), and lactate dehydrogenase (LDH, cat. no. 20220422) assay kits were obtained from the Nanjing Jiancheng Bioengineering Institute, Nanjing, China. Methylthiazolyl tetrazolium bromide (MTT) was purchased from Sigma-Aldrich, St. Louis, MO, USA. The BCA Protein Assay Kit, Radioimmunoprecipitation Assay (RIPA) Lysis Buffer, and TriQuick Total RNA Extraction Reagents were purchased from Beijing Solarbio Science & Technology Co., Ltd., Beijing, China. The cDNA reverse transcription kit was purchased from Nanjing Vazyme Biotech Co., Ltd., Nanjing, China.

### 2.2. Animals and Ethics Statement

Sixty healthy male BALB/c mice, weighing 20 ± 2 g, were provided by the Experimental Animal Center of Tongji Medical College at the Huazhong University of Science and Technology. The animal license number is SCXK (Hubei) 2010-0007. The Experimental Animal Ethics Committee of Henan University of Science and Technology granted approval for the animal-related experimental protocols (approval no. 20200519). All experiments involving animals were carried out in accordance with the National Act on the Use of Experimental Animals in China, with appropriate measures implemented to minimize both animal usage and any associated distress.

### 2.3. Cell Culture

RAW 264.7 macrophage cell lines were donated by the Chinese Academy of Medical Sciences. The cells were preserved and subcultured by our laboratory, cultured in RPMI medium 1640 (Beijing Solarbio Science & Technology Co., Ltd., Beijing, China), supplemented with 10% fetal bovine serum (Gibco, Thermo Fisher Scientific, Inc., Waltham, MA, USA), and incubated at 37 °C in a 5% CO_2_, humidified atmosphere. The cells were digested with trypsin (SIGMA, St. Louis, MO, USA).

### 2.4. Cell Viability Assay

RAW 264.7 macrophages were seeded into 96-well plates at a density of 4 × 10^3^ cells/well and cultured for 24 h. Subsequently, the cells were treated with LPS (1 μg/mL), various concentrations of SS-b2 (0, 0.25, 0.5, 1, 2, 4, 8, 16, 32, 64, and 128 μg/mL), and DEX (1 μg/mL). Different concentrations of SS-b2 and DEX were initially cultured for 1 h, followed by co-culturing with LPS (1 μg/mL) for an additional 20 h. Afterward, 20 μL of MTT solution (5 mg/mL) was added to each well and incubated for an additional 4 h at 37 °C. The culture medium was then replaced with 200 µL of DMSO, and an ELx800 microplate reader (Bio-Tek, Winooski, VT, USA) was used to measure the absorbance (570 nm) of each well. The data presented here are the results of at least three independent experiments.

### 2.5. Nitrite Detection

RAW 264.7 macrophages were treated with SS-b2 and/or LPS for 24 h. Nitrite concentration in the culture supernatant was estimated using the Griess reaction, as described by Kim et al. [22]. A standard curve was established using NaNO_2_. 

### 2.6. Establishment of Primary Liver Cancer Model in Mice

We employed DEN-induced mice to establish the primary liver cancer (PLC) model [23]. The mice were randomly divided into six groups (*n* = 10) as follows: control, model (DEN at 50 mg/kg), SS-b2 (SS-b2 at 1.5, 3, and 6 mg/kg), and DOX (DOX at 1 mg/kg). For the initial four weeks, excluding the control group, the mice received intraperitoneal injections (ip) of DEN at a dose of 50 mg/kg twice a week, which was subsequently reduced to once a week from the 5th to the 19th week. The control group was given an equivalent volume of normal saline. Starting from the fifth week, the SS-b2 groups received intraperitoneal injections of SS-b2 at 1.5, 3, and 6 mg/kg doses once a day. The DOX group received intraperitoneal injections of DOX at a dose of 1 mg/kg once every two days. All treatments spanned 19 weeks. Twenty-four hours after the final administration, the mice were weighed, and blood samples were collected. Additionally, hepatic tissues were carefully isolated and processed for further analysis. The liver index was calculated using the following formula: Liver index (g/kg) = liver mass (g)/body mass (kg). Surface liver nodules were compared and recorded. Some liver tissues were fixed in 4% paraformaldehyde, while the remainder was frozen in a −80 °C ultra-low temperature freezer for the detection of signaling pathway proteins.

### 2.7. Serum Liver Function Test

Blood samples were centrifuged at 3000× *g* for 10 min at 4 °C, and the supernatant was collected. Serum levels of ALT, AST, LDH, and alpha-fetoprotein (AFP) were determined following the instructions provided with the respective kits.

### 2.8. Hematoxylin and Eosin Staining (H&E)

Liver tissues from the mice were fixed in 4% paraformaldehyde for 12 h at room temperature and then rinsed for 24 h. Subsequently, the fixed liver tissues were dehydrated through a series of ethanol concentrations, including 75%, 80%, 95%, and two rounds of 100% ethanol, with each step lasting 30 min at room temperature. For clearing, the dehydrated tissues were immersed in xylene twice, each time for 30 min at room temperature. Afterward, the liver tissues were embedded in paraffin and sectioned to a thickness of 5 µm. These sections were subjected to hematoxylin and eosin (H&E) staining (Nanjing Jiancheng Bioengineering Institute, Nanjing, China) for histologic examination and subsequently observed and photographed under an optical microscope to assess liver injury.

### 2.9. Immunohistochemistry Staining

Paraffin sections of the liver tissues were utilized for immunohistochemical staining. They were first deparaffinized and rehydrated following the instructions provided with the immunohistochemical kit (Solarbio, Beijing, China), which included antigen retrieval. Subsequently, the samples were washed with PBS and then incubated in a 5% BSA blocking solution [24]. The primary antibodies used were as follows: anti-rabbit Ki67 (1:250 dilution, Abcam, Cambridge, UK), anti-rabbit STK4 (1:200 dilution, Abcam, Cambridge, UK), and anti-rabbit IRAK1 (1:50 antibody, Wuhan Proteintech Group, Inc., Wuhan, China). Afterward, the samples were incubated overnight at 4 °C. After washing, the secondary antibody (goat anti-rabbit IgG, Wuhan Proteintech Group, Inc., Wuhan, China) was applied, and the samples were incubated in the dark for 30 min at 37 °C. Finally, the sections were sealed with a neutral gum, photographed, and observed with a microscope.

### 2.10. Quantitative Real-Time Polymerase Chain Reaction (qPCR)

For total RNA extraction, TriQuick Total RNA Extraction Reagents were used following the manufacturer’s instructions. Peripheral blood mononuclear cells were isolated from fresh mouse whole blood and separated using leukocyte separation solution. Then, 0.4 mL of whole blood was added to 7.6 mL of leukocyte separation solution, and the sample was centrifuged at 1500× *g* for 10 min at 4 °C. The intermediate leukocyte layer was carefully transferred into a new centrifuge tube. The leukocyte precipitation was resuspended in 5 mL PBS, centrifuged at 1500× *g* for 5 min, and the supernatant was discarded. Then, the wash was repeated with PBS and the precipitation was retained. The supernatant was discarded, and 1 mL of TriQuick Total RNA Extraction Reagent was added to the precipitate. The RNA concentration was determined using spectrophotometry (NanoDrop2000). Subsequently, total RNA was reverse-transcribed into cDNA, followed by quantitative real-time polymerase chain reaction (qPCR) in accordance with the manufacturer’s instructions. GAPDH served as a reference gene, and the relative levels of the target genes were calculated using the 2^−ΔΔCT^ method [25]. Primer sequences for mouse STK4, IRAK1, IL-1β, IL-6, and TNF-α were designed based on GenBank gene information (Table 1).

### 2.11. HepG2 Cell Culture and siRNA Transfection

HepG2 cells were seeded into 96-well plates at a density of 4 × 10^3^ cells/well and cultured for 24 h. Subsequently, the cells were treated with various concentrations of SS-b2 (0, 25, 50, 100, 200, 400 μg/mL) and cell viability was assessed using the MTT assay. For the silencing of human *STK4*, siRNA sequences were designed and synthesized by GenePharma Co., Ltd. (Shanghai, China). The siRNA sequences were as follows: sense 5′-GAGCUAUGGUCAGAUAACU-3′ and antisense 5′-AGUUAUCUGACCAUAGCUCTT-3′ for STK4. HepG2 cells, at 50% confluence, were transfected with 20 µM *STK4* siRNA using Lipofectamine^TM^ 3000 Transfection Reagent (Invitrogen, Waltham, MA, USA), following the manufacturer’s instructions. The cells were initially seeded a day before transfection, using Dulbecco’s Modified Eagle’s Medium (DMEM, Solarbio, Beijing, China) supplemented with 10% fetal bovine serum (FPS, Gibco, Waltham, MA, USA). After 48 h, the cells were subjected to RT-PCR and Western blotting analysis.

### 2.12. Western Blot Analysisblot Analysis

Liver tissues were lysed using RIPA lysis buffer, and the protein concentration was determined with a BCA protein assay kit. Total proteins (30 μg) were separated by 10% SDS-PAGE. Then, the proteins were transferred onto a PVDF membrane, which was blocked with 5% skimmed milk powder and incubated in a 37 °C incubator for 1 h. Primary antibodies against STK4 (cat. no. ab265442; dilution 1:10,000, Abcam, Cambridge, UK), IRAK1 (1:1000), phospho-NF-κB p65 (cat. no. ab76302; dilution 1:1000, Abcam), and β-actin (cat. no. 60008-1-1 g; dilution 1:10,000, Wuhan Proteintech Group, Inc., Wuhan, China) were added and incubated overnight at 4 °C. Following this, goat anti-rabbit IgG secondary antibody (cat. no. ZB-2301; 1:6000; Wuhan Proteintech Group, Inc., Wuhan, China) and goat anti-mouse IgG secondary antibody (cat. no. ZB-2305; 1:1000; Wuhan Proteintech Group, Inc., Wuhan, China) were added and incubated at 37 °C for 1 h. The protein bands were semi-quantitatively analyzed using Gel-Pro Analyzer 4.0. The relative expression level of the target protein was expressed as the ratio of the integral absorbance of the target protein band to the internal reference protein band.

### 2.13. Statistical Analysis

Quantitative data were expressed as mean ± standard deviation (SD). We performed a statistical analysis using GraphPad Prism 7.0 software. Multiple group means were compared using Tukey’s test, and two-group means were compared using Student’s *t*-test. A significance level of *p* < 0.05 was considered statistically significant.

## 3. Results

### 3.1. Effect of Saikosaponin-b2 on Liver Function in Primary Liver Cancer Mice

To evaluate the effect of SS-b2 on liver function in PLC mice in vivo, we examined the levels of AST, ALT, and LDH in the serum of PLC mice treated with 1.5, 3, or 6 mg/kg of SS-b2 or 1 mg/kg of DOX, over a 15-week period. As shown in Figure 1A–C, the serum levels of AST, ALT, and LDH in the model group were significantly higher than those in the control group (*p* < 0.01). Compared to the model group, both the SS-b2 and positive drug (DOX) groups exhibited a significant reduction in the serum levels of AST, ALT, and LDH (*p* < 0.05 and *p* < 0.01, respectively). These findings suggested that SS-b2 provides protection against DEN-induced liver injury, leading to a notable decrease in transaminase levels and an improvement in liver function among PLC mice.

### 3.2. Antitumor Effect of Saikosaponin-b2 in Primary Liver Cancer Mice

As shown in Figure 1E, the liver surfaces of mice in the model group were predominantly covered with different sizes of nodules compared to the control group. In the SS-b2 group, the number of nodules on the liver surface decreased with an increase in the SS-b2 dosage. Additionally, the liver index of the SS-b2 group was significantly lower than that of the model group (*p* < 0.01, Figure 1D). H&E staining was performed to evaluate pathomorphological changes in liver tissues, with the results shown in Figure 1F. The liver tissues of mice in the model group exhibited typical structures of columnar and glandular duct-like cancer nests. These cancer cells were arranged in stripes or clusters and infiltrated the surrounding liver tissues with considerable inflammatory cell infiltration. Generally, the liver tissue exhibited a significantly heightened degree of malignancy (Figure 1J). Following treatment, both the SS-b2 and DOX groups showed a significant reduction in cancer cell proliferation, the number of cancer nests, and the degree of cancerous lesions (Figure 1F,J). Furthermore, they exhibited reduced infiltration of inflammatory cells. Alterations in serum AFP levels (Figure 1I) showed a significant increase in the model group (*p* < 0.01) and a significant decrease in the treated groups (*p* < 0.01). These results indicated that SS-b2 effectively inhibited the development of liver cancer in DEN-induced PLC mice.

### 3.3. Effect of Saikosaponin-b2 on Ki67 in Primary Liver Cancer Mice

To evaluate the impact of SS-b2 treatment on PLC development, Ki67 was assessed using immunohistochemistry with an anti-Ki67 antibody, as shown in Figure 1G. Ki67-positive staining was significantly higher in the liver tissues of the model group than in the control group. However, compared to the model group, both the SS-b2 and DOX groups showed a significant reduction in positively stained areas for Ki67 and the number of Ki67-positive cells (*p* < 0.01, Figure 1H). Considering these observations, we can conclude that SS-b2 effectively suppresses the malignant proliferation of liver cancer cells.

### 3.4. Effect of Saikosaponin-b2 on the Expressions of STK4 and IRAK1 in Primary Liver Cancer Mice

The immunohistochemical results (Figure 2A,B) showed that the positive expression of STK4 increased, while the relative positive expression of IRAK1 significantly declined in the liver tissues of the model group (*p* < 0.01, Figure 2C). Treatment with SS-b2 and DOX significantly increased the positive expression of STK4 while decreasing the IRAK1 expression (*p* < 0.01, Figure 2D). These findings showed that SS-b2 inhibits PLC development in mice by increasing STK4 expression and decreasing IRAK1 expression. Similarly, the mRNA expression levels of *STK4* and *IRAK1* were analyzed in peripheral blood mononuclear cells using qPCR (Figure 2E,F). Compared to the control group, the mRNA expression of *STK4* increased, while that of *IRAK1* decreased significantly in the peripheral blood mononuclear cells of the model group (*p* < 0.01). Additionally, a negative correlation was observed between the mRNA expression levels of *STK4* and *IRAK1* in the macrophages of all mouse groups (Figure 2G). These findings suggested that SS-b2 suppresses PLC development by targeting STK4 and IRAK1.

### 3.5. Effect of Saikosaponin-b2 on STK4 Expression and IRAK1/NF-κB Signaling Axis In Vivo

Figure 3A,B showed the results of the Western blot analysis. Compared to the control group, the liver tissues of mice in the model group exhibited significantly decreased protein expression of STK4, alongside substantial increases in the protein levels of IRAK1 and phospho-NF-κB p65 (*p* < 0.01). These findings suggested that the expression of STK4 was decreased and IRAK1/NF-κB signaling axis was activated in the livers of DEN-induced PLC mice. Compared with the model group, the SS-b2 and DOX treatments significantly increased the protein expression of STK4 while significantly decreasing the elevated levels of IRAK1 and phospho-NF-κB p65 proteins in the liver tissues of PLC mice (*p* < 0.01). Moreover, Figure 3C–E illustrated the mRNA expression of relevant inflammatory factors in the liver tissues of the mice, as assessed through qPCR. The model group showed significantly higher mRNA levels of *IL-1β*, *IL-6*, and *TNF-α* than the normal group. Upon SS-b2 and DOX treatment, the mRNA expression levels of *IL-1β*, *IL-6*, and *TNF-α* were significantly reduced in liver tissues (*p* < 0.01). These results suggested that SS-b2 may prevent the malignant development of PLC by negatively regulating the IRAK1/NF-κB signaling axis through up-regulating STK4, thereby decreased the levels of proinflammatory cytokines.

### 3.6. Effect of Saikosaponin-b2 on the Expressions of STK4, IRAK1 and NF-κB Proteins in HepG2 Cells

HepG2 liver cancer cells were utilized to investigate the potential therapeutic impact of SS-b2 on primary liver cancer in vitro. Subsequently, the antitumor effects of SS-b2 were examined. The viability of HepG2 cells was assessed using MTT assay. As presented in Figure 4A, SS-b2 significantly inhibited HepG2 liver cancer cell proliferation. Based on these results, 15, 30, and 60 µg/mL SS-b2 were selected for the subsequent experiments. To further assess the effect of SS-b2 on the IRAK1/NF-κB signaling axis through targeting STK4, the protein expression levels of STK4, IRAK1, and phospho-NF-κB p65 were analyzed. As presented in Figure 4E,F, with an increase in SS-b2 concentration, the protein expression level of STK4 was upregulated, while that of IRAK1 and phospho-NF-κB p65 were downregulated (*p* < 0.01).

### 3.7. Effects of Saikosaponin-b2 on the IRAK1/NF-κB Signaling Axis by Targeting STK4

We employed siRNA to effectively silence the expression of *STK4* in HepG2 liver cancer cells to investigate the effect of *STK4* gene knockdown on the anti-tumor effect of SS-b2 in vitro. We utilized both RT-PCR and Western blot techniques to measure the effects of SS-b2 on regulating STK4 and the IRAK1/NF-κB signaling axis, as well as the influence of *STK4* siRNA on this pathway. RT-PCR and Western blot results were shown in Figure 4; the expression of IRAK1 and NF-κB were elevated following *STK4* knockdown compared with the negative control group. These results suggested that *STK4* knockdown upregulates IRAK1 expression and thus the activation of NF-κB activity revealed by the increase in the levels of proinflammatory cytokines, consequently impairing SS-b2-induced inhibition of liver cancer development. Our findings further supported the notion that SS-b2 prevented PLC by up-regulating STK4, which negatively regulated the IRAK1/NF-κB signaling axis.

### 3.8. Effect of Saikosaponin-b2 on LPS-Induced Viability and Nitric Oxide Secretion in RAW 264.7 Macrophages

RAW 264.7 macrophage cells were used to investigate the anti-inflammatory effect of SS-b2 in vitro. To evaluate the effects of SS-b2 on RAW 264.7 macrophages, MTT assay was performed to determine the nontoxic concentration of SSb2 in RAW 264.7 macrophages for 24 h. As presented in Figure 4A, SSB2 did not show obvious cytotoxic effects at concentrations of 2.5–160 μg/mL (*p* < 0.01). Griess assay was performed to assess the inhibitory effects of SSB2 on NO release induced by LPS, cells were pretreated with SSB2 for 1 h and then stimulated with LPS (1 μg/mL) for 24 h. Cells that were LPS-stimulated (1 μg/mL) highly released NO compared with control cells (*p* < 0.01, Figure 4A). However, SSB2 significantly suppressed LPS-induced NO production (*p* < 0.01). Then, the cells were pretreated with SS-b2 concentrations of 15, 30, and 60 µg/mL and DEX of 1 μg/mL, and then stimulated with LPS of 1 μg/mL for 6, 12, and 24 h. The results showed that SS-b2 and DOX had the best inhibitory effect when stimulated with LPS for 24 h. Based on these results, macrophages were stimulated with 1 μg/mL of LPS for 24 h as the modeling condition. And SS-b2 concentrations of 15, 30, and 60 µg/mL were selected for therapeutic doses. The subsequent experiments were conducted in six groups as follows: control group, LPS group (treated with 1 μg/mL of LPS), SS-b2 group (treated with 15, 30, and 60 μg/mL of SS-b2), and DEX group (treated with 1 μg/mL of DEX).

### 3.9. Anti-Inflammatory Effect of Saikosaponin-b2 on LPS-Stimulated RAW 264.7 Macrophages

To further elucidate the anti-inflammatory function of SS-b2, qPCR was used to detect the mRNA expression levels of *IL-1β*, *IL-6*, and *TNF-α* in macrophages (Figure 5C–E). After protective pretreatment with SS-b2 (15, 30, and 60 μg/mL) and DEX (1 μg/mL) for 1 h, the cells were exposed to LPS (1 μg/mL) for 6 h. Compared to the control group, the mRNA expression levels of *IL-1β*, *IL-6*, and *TNF-α* significantly increased in the LPS group but significantly decreased in the SS-b2 and DEX groups (*p* < 0.01). The changes in macrophage morphology were observed microscopically and are presented in Figure 4E. Compared to the LPS group, the SS-b2 and DEX treatments significantly reduced macrophage cell numbers and inhibited cell proliferation. Furthermore, most of the differentiated cells regained their characteristic round or oval shape, showing that SS-b2 decreased the secretion of various inflammatory factors (e.g., NO, IL-1β, IL-6, and TNF-α) by RAW 264.7 macrophages upon LPS stimulation in vitro. Additionally, SS-b2 inhibited the proliferation and differentiation of LPS-stimulated RAW 264.7 macrophages, thereby reducing the inflammatory response.

### 3.10. Effect of SS-b2 on the Expressions of STK4, IRAK1, and NF-κB Proteins in LPS-Stimulated RAW 264.7 Macrophages

To evaluate the effects of SS-b2 on the expressions of STK4, IRAK1, and NF-κB in LPS-stimulated RAW 264.7 macrophages, western blotting was used to assess the expression levels of relevant proteins in RAW 264.7 macrophages. In the LPS model group, the protein expression levels of STK4 significantly decreased, while those of IRAK1 and phospho-NF-κB p65 significantly increased (*p* < 0.01, Figure 6). In addition, with an increase in SS-b2 concentration, the protein expression level of STK4 was upregulated, while that of IRAK1 and phospho-NF-κB p65 were downregulated in the SS-b2 groups (*p* < 0.05, Figure 6). This result indicated that SS-b2 can suppress the expression of downstream transcription factors by regulating STK4 and IRAK1/NF-κB signaling axis in LPS-stimulated RAW 264.7 macrophages, further verifying that SS-b2 can inhibit PLC development through upregulating STK4 to suppress the IRAK1/NF-κB signaling axis.

## 4. Discussion

The relationship between chronic inflammation and tumors has become crucial in PLC treatment. Studies have shown that approximately 90% of PLCs are associated with long-term inflammatory responses and persistent liver injury [26]. Prolonged inflammation can cause gene mutations that induce PLC. It can also cause immune system dysfunction, altering the liver microenvironment and fostering tumor cell growth [27,28]. Our previous study demonstrated that SS-b2 has remarkable hepatoprotective effects and inhibits tumor growth in H22-transplanted mice with HCC [18]. Based on these previous findings, we aimed to investigate whether SS-b2 inhibits PLC development, particularly inflammation-related liver cancer, by targeting STK4/IRAK1 through in vivo and in vitro experiments. We seek to provide experimental evidence supporting the use of SS-b2 as a novel monomodal drug targeting STK4/IRAK1 for PLC treatment. 

In this study, we found that SS-b2 significantly reduced the levels of ALT, AST, and LDH in the livers of DEN-induced PLC mice, thereby improving liver function. Additionally, SS-b2 decreased the number of surface nodules, the size of cancer nests, and the number of cancer cells in liver tissues, improving cancerous lesions in mice with PLC. Our findings indicated that SS-b2 exerted certain antitumor effects on PLC mice. Furthermore, SS-b2 has been shown to decrease the secretion of various inflammatory factors, such as NO, IL-1β, IL-6, and TNF-α, by RAW 264.7 macrophages upon LPS stimulation in vitro, confirming its anti-inflammatory effects.

Ki67, an antigen associated with cell proliferation, is highly expressed in most malignant cells and rarely detected in normal cells. Therefore, it is a key indicator of tumor malignancy [29]. According to liver tissue immunohistochemistry, SS-b2 markedly decreased Ki67 expression and the quantity of Ki67-positive cells in liver tissue, indicating antitumor activity. AFP, a serum marker clinically used for PLC diagnosis and highly expressed in patients with liver cancer [30], exhibited a significant reduction following SS-b2 treatment. 

To further investigate the mechanisms underlying the antitumor and anti-inflammatory effects of SS-b2 on PLC, we investigated the influence of SS-b2 on STK4 regulation to impact the IRAK4/NF-κB signaling axis. Chronic liver inflammation and oncogene inactivation are important factors in PLC development [31,32]. Clinical studies have found substantial macrophage infiltration in the tumor tissues of patients with liver cancer, as well as a negative correlation between low STK4 levels and high IRAK1 levels within intracellular or peripheral blood mononuclear cells [11]. Employing IRAK1 inhibitors resulted in a significant reduction in chronic inflammation and HCC caused by *STK4* deficiency [33]. Studies have revealed the expression of STK4 in immune and tumor cells [34,35]. Moreover, STK4 regulates the transcriptional activity of YAP/TAZ in the Hippo pathway, inhibits the proliferation of liver cancer cells [36], and degrades in macrophages by binding and phosphorylating IRAK1. Consequently, it blocks the activation of the inflammatory pathway-associated factor NF-κB, along with its downstream proinflammatory factors and antiapoptotic gene expression, impacting the progression of liver cancer [37,38]. Our study findings demonstrated that SS-b2 markedly upregulated STK4 expression and downregulated IRAK1 expression in the liver tissues and peripheral blood of mice with PLC. Similarly, SS-b2 significantly decreased the protein expression levels of NF-κB, as well as the expression of inflammatory factors (IL-1β, IL-6, and TNF-α). The siRNA results showed that *STK4* knockdown upregulated the expression of IRAK1 and thus the activation of NF-κB activity, consequently impairing SS-b2-induced inhibition of liver cancer development. Therefore, SS-b2 could potentially inhibit downstream molecules of the NF-κB pathway, including IL-1β, IL-6, and TNF-α, by increasing STK4 protein levels and decreasing IRAK1 and NF-κB protein expression. 

In this study, we found that SS-b2 exerted antitumor effects by upregulating the STK4-negative regulation of the IRAK1/NF-kB signaling axis in vivo and in vitro experiments. The results of this study also showed that SS-b2 significantly decreased the mRNA expression levels of *IL-1β*, *IL-6*, and *TNF-α* in both RAW 264.7 macrophages in vitro and liver tissues of DEN-induced model mice in vivo. This suggested that SS-b2 had the potential to modulate the production of these pro-inflammatory factors and attenuate persistent inflammatory responses. Pro-inflammatory cytokines play a critical role in initiating and regulating immune responses. They can trigger inflammatory immune responses and chemokine production, which recruit additional immune cells to the site of injury or infection [39,40]. IL-1β, IL-6, and TNF-α are key mediators of the inflammatory and immune responses. They play crucial roles in processes such as viral clearance and antitumor immune responses. 

Although inflammation is generally considered as a tumor-promoting factor, some studies suggested that inflammatory cell infiltration into tumors may be associated with better prognosis. When driven by tumor-specific Th1 cells, inflammation can prevent cancer. In the Th1 environment, pro-inflammatory cytokines IL-1α, IL-1β, and IL-6 may participate in cancer eradication by recruiting white blood cells from the circulation and stimulating cells. On the other hand, inflammation that lacks a sufficient number of tumor-specific Th1 cells or involves other types of immune cells may not have the same protective effect [41]. It is best to avoid using drugs that may inhibit the inflammatory immune response driven by tumor-suppressive Th1. This study suggested that SS-b2 may impact the development of liver cancer by inhibiting the activation or production of pro-inflammatory factors, attenuating persistent inflammatory responses, and preventing immune dysfunction. These results did not involve in Th1 cells activation and possibly was related to its direct attenuate inflammation and preventing the progression of hepatitis to cancer. These findings provided a basis for further exploration of SS-b2 as a potential therapeutic agent for liver cancer and related inflammatory conditions. 

In addition, liver cancer is closely associated with immune cell infiltration. *STK4* is a key tumor suppressor gene in HCC and is normally expressed in immune cells. IRAK1 and NF-κB play key roles in the production of pro-inflammatory cytokines in macrophages as well as in cell proliferation in liver tumor cells. Macrophages are the most abundant immune cells in the tumor microenvironment and are involved in every stage of tumorigenesis and progression, playing a critical role in initiating and maintaining inflammation, fibrosis, and cancer progression [42,43]. Macrophages residing in the tumor microenvironment are known as tumor-associated macrophages, which are central regulators in the tumor microenvironment and are not only closely associated with HCC initiation and progression, but also involved in multidrug resistance (including immunotherapy) in HCC, which affects patient survival and prognosis. Our results demonstrate that SS-b2 upregulates STK4 expression in LPS-induced macrophages, resulting in negative regulation of the IRAK1/NF-KB signaling pathway. This leads to inhibition of the release of downstream pro-inflammatory cytokines, ultimately suppressing carcinogenesis that results from persistent inflammatory microenvironmental factors.

In summary, we demonstrated the potential of SS-b2 to inhibit PLC by altering the expression of STK4/IRAK1 targets, thereby suppressing the activation of the downstream NF-κB inflammatory signaling pathway and subsequent production of inflammatory factors. Our findings suggested that SS-b2 could serve as a new drug targeting STK4/IRAK1 for PLC treatment.

## Figures and Tables

**Figure 1 biomedicines-11-02859-f001:**
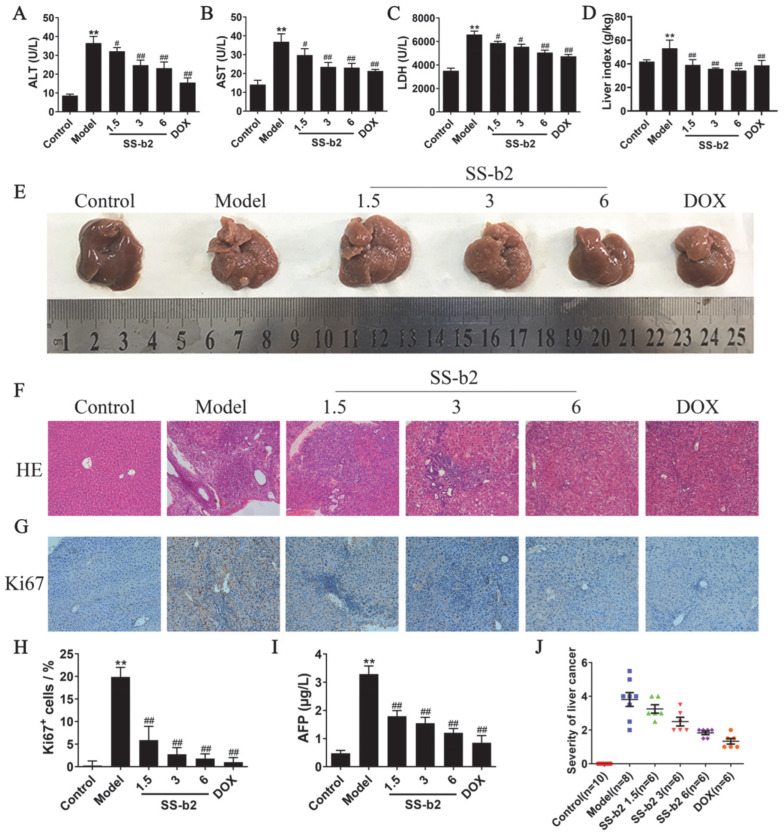
Effects of saikosaponin-b2 on primary liver cancer mice. (**A**–**C**) Statistical analysis of the serum transaminase levels of mice in each group. (**D**) Liver index of the mice in each group. (**E**) Representative images of the liver tissues in each group. (**F**) Pathomorphological changes in liver tissues of primary liver cancer mice (H&E staining, magnification ×200). (**G**) Representative images of Ki67 immunohistochemical staining of liver tissues in each group (magnification ×200). (**H**) Semiquantitative analysis of Ki67 immunohistochemical staining of liver tissues in each group. (**I**) Statistical analysis of the serum AFP levels of mice in each group. (**J**) Degree of malignancy in primary liver cancer mice. Data are presented as mean ± SD of three independent experiments and differences between mean values were assessed by Student’s *t*-test. ** *p* < 0.01, compared with the control group; # *p* < 0.01, ## *p* < 0.01, compared with the model group. SS-b2, saikosaponin-b2; DOX, doxorubicin; H&E, hematoxylin, and eosin.

**Figure 2 biomedicines-11-02859-f002:**
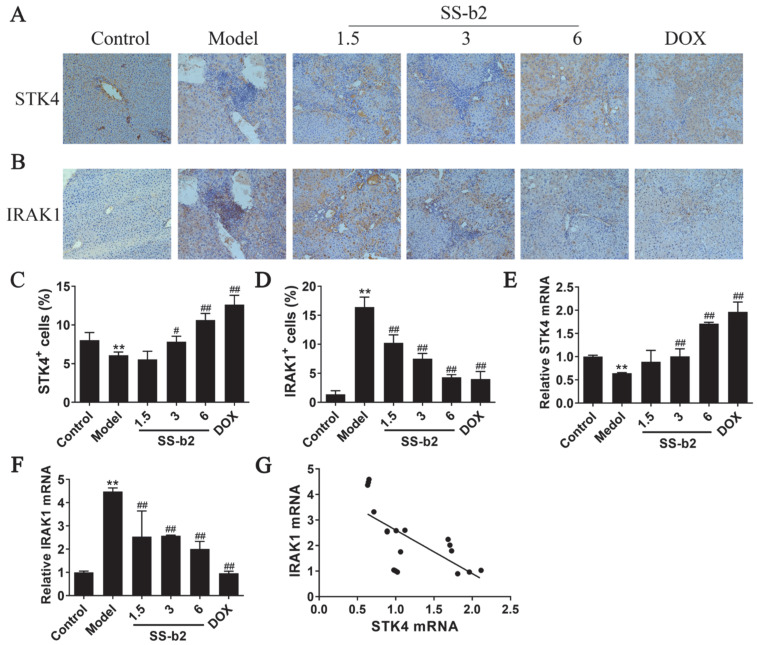
Effects of saikosaponin-b2 on the expression of STK4 and IRAK1 in primary liver cancer mice. (**A**,**B**) Representative images of STK4 and IRAK1 immunohistochemical staining of liver tissues in each group (magnification ×200). (**C**,**D**) Semiquantitative analysis of STK4 and IRAK1 immunohistochemical staining of liver tissues in each group. (**E**,**F**) Statistical analysis of the mRNA expression levels of *STK4* and *IRAK1* in the peripheral blood mononuclear cells of mice in each group. (**G**) Correlation analysis of the mRNA expression levels of *STK4* and *IRAK1* in the peripheral blood mononuclear cells of mice in each group. Data are presented as mean ± SD of three independent experiments and differences between mean values were assessed by Student’s *t*-test. ** *p* < 0.01, compared with the control group; # *p* < 0.01, ## *p* < 0.01, compared with the model group. SS-b2, saikosaponin-b2; DOX, doxorubicin.

**Figure 3 biomedicines-11-02859-f003:**
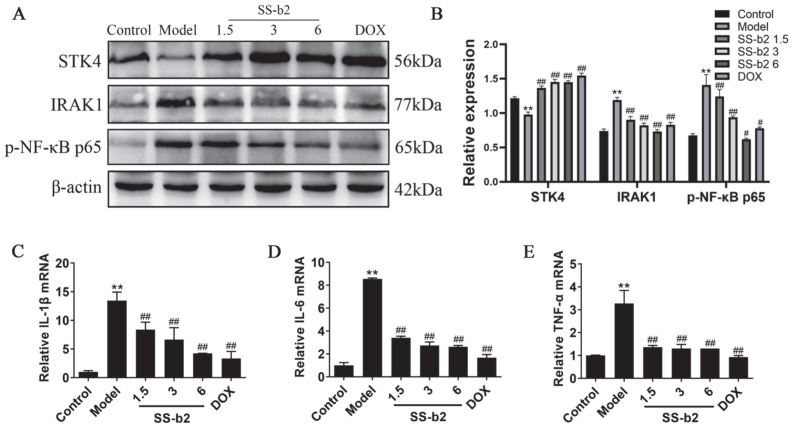
Effects of saikosaponin-b2 on the STK4/IRAK1/NF-κB pathway in vivo. (**A**) Western blot bands for the expression of STK4, IRAK1, and phospho-NF-κB p65. (**B**) Semiquantitative analysis of STK4, IRAK1, and p-NF-κB p65. (**C**–**E**) Statistical analysis of the mRNA expression levels of *IL-1β*, *IL-6*, and *TNF-α* in the liver tissues of mice in each group. Mice in each group were treated with DEN (50 mg/kg), different concentrations of SS-b2 (1.5, 3, and 6 mg/kg) and DOX (1 mg/kg), and protein expression in liver tissues were detected using Western blotting. β-actin served as a loading control. Data are presented as mean ± SD of three independent experiments and differences between mean values were assessed by Student’s *t*-test. ** *p* < 0.01, compared with the control group; # *p* < 0.01, ## *p* < 0.01, compared with the model group. SS-b2, saikosaponin-b2; DOX, doxorubicin.

**Figure 4 biomedicines-11-02859-f004:**
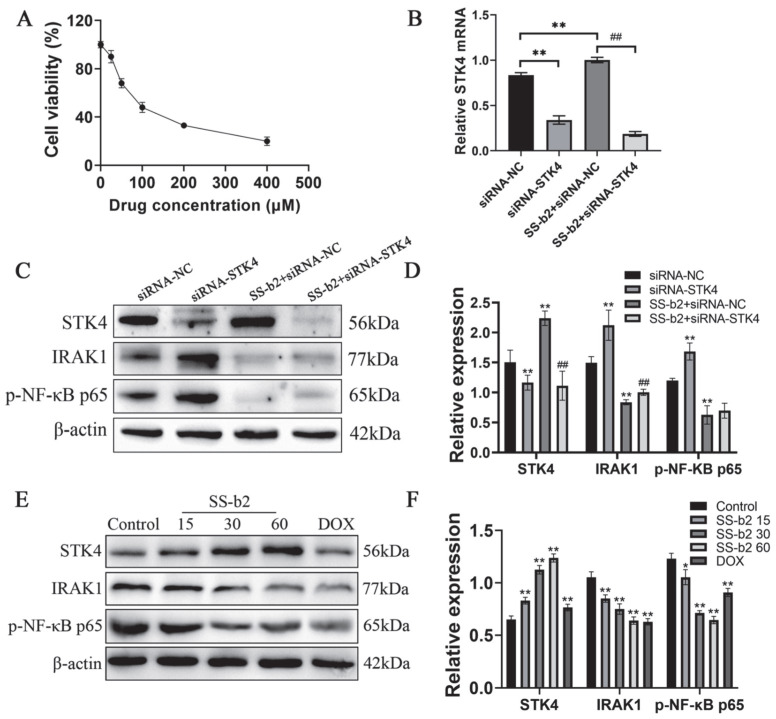
Effects of saikosaponin-b2 on the IRAK1/NF-κB signaling axis by targeting STK4 in HepG2 cells. (**A**) Effect of saikosaponin-b2 on the cell viability of HepG2. Cells were treated with various doses of SS-b2 for 24 h, and cell viability was measured using the MTT assay. (**B**) The mRNA expression levels of *STK4* siRNA in HepG2 were detected by RT-PCR. ** *p* < 0.01, compared with siRNA-NC; ## *p* < 0.01, compared with SS-b2 + siRNA-NC. (**C**) Western blot bands for the expression of STK4, IRAK1, and p-NF-κB p65 after silencing STK4 expression. (**D**) Semiquantitative analysis of STK4, IRAK1, and p-NF-κB p65. ** *p* < 0.01, compared with siRNA-NC; ## *p* < 0.01, compared with SS-b2 + siRNA-NC. (**E**) Western blot bands for the expression of STK4, IRAK1, and p-NF-κB p65. HepG2 in each group were treated with different doses of SS-b2 (15, 30, and 60 μg/mL) and DOX (2 μg/mL), β-actin served as a loading control. (**F**) Semiquantitative analysis of STK4, IRAK1, and p-NF-κB p65. * *p* < 0.05, ** *p* < 0.01, compared with control group. Data are presented as mean ± SD of three independent experiments and differences between mean values were assessed by Student’s *t*-test. SS-b2, saikosaponin-b2; DOX, doxorubicin.

**Figure 5 biomedicines-11-02859-f005:**
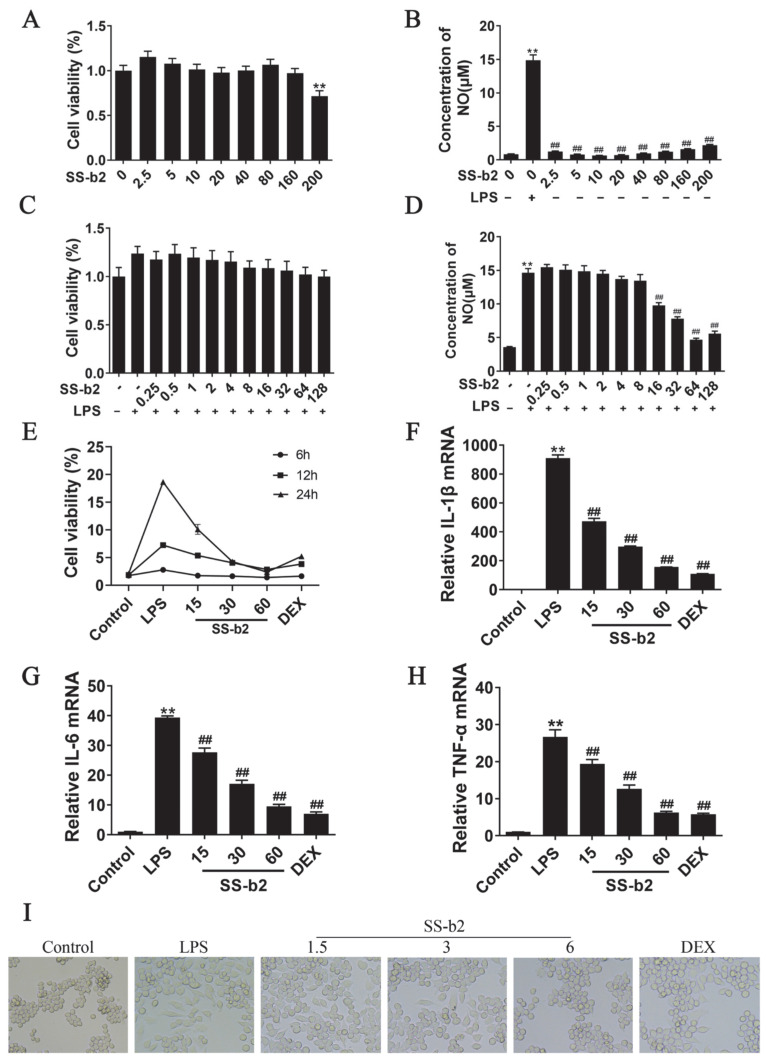
Anti-inflammatory effects of saikosaponin-b2 on LPS-stimulated RAW 264.7 macrophages. (**A**) Effect of saikosaponin-b2 on the cell viability of RAW 264.7 macrophages. Cells were treated with various doses of SS-b2 (0–200 μg/mL) for 24 h, and cell viability was measured using the MTT assay. (**B**) Effect of saikosaponin-b2 on NO content in RAW 264.7 macrophages. Cells were treated with various doses of SS-b2 (0–200 μg/mL) for 24 h, and the NO content was measured using the Griess method. (**C**) Effect of saikosaponin-b2 on the viability of LPS-stimulated RAW 264.7 macrophages. Cells were pretreated with various doses of SS-b2 (0.25–128 μg/mL) for 1 h and then incubated with 1 μg/mL of LPS for 24 h. Cell viability was measured using the MTT assay. (**D**) Effect of saikosaponin-b2 on NO content in LPS-stimulated RAW 264.7 macrophages. Cells were pretreated with various doses of SS-b2 (0.25–128 μg/mL) for 1 h and incubated with 1 μg/mL of LPS for 24 h. NO content was measured using the Griess method. (**E**) Cells were pretreated with SS-b2 concentrations of 15, 30, and 60 µg/mL and DEX of 1 μg/mL, and then stimulated with LPS of 1 μg/mL for 6, 12, and 24 h. (**F**–**H**) The mRNA expression levels of *IL-1β*, *IL-6*, and *TNF-α* in macrophages were detected using qPCR. (**I**) Effect of SS-b2 on LPS-stimulated RAW 264.7 macrophages (magnification ×100). Data are presented as mean ± SD of three independent experiments and differences between mean values were assessed by Student’s *t*-test. ** *p* < 0.01, compared with the control group; ## *p* < 0.01, compared with the model group. SS-b2, saikosaponin-b2; DEX, dexamethasone.

**Figure 6 biomedicines-11-02859-f006:**
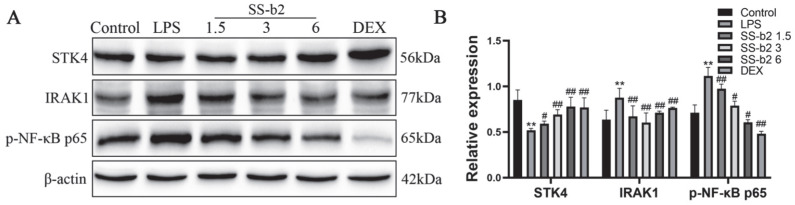
Effects of saikosaponin-b2 on the STK4/IRAK1/NF-κB pathway in LPS-stimulated RAW 264.7 macrophages. (**A**) Western blot bands for the expression of STK4, IRAK1, and p-NF-κB p65. (**B**) Semiquantitative analysis of STK4, IRAK1, and p-NF-κB p65. Macrophages in each group were treated with LPS (1 μg/mL), different concentrations of SS-b2 (15, 30, and 60 μg/mL) and DEX (1 μg/mL); β-actin served as a loading control. Data are presented as mean ± SD of three independent experiments, and differences between mean values were assessed by Student’s *t*-test. ** *p* < 0.01, compared with the control group; # *p* < 0.01, ## *p* < 0.01, compared with the model group. SS-b2, saikosaponin-b2; DEX, dexamethasone.

**Table 1 biomedicines-11-02859-t001:** Primer names and sequences.

Primer Names	Sequences (5′–3′)
IL-1β	F: TCTCGCAGCAGCACATCAAC
	R: ACCAGCAGGTTATCATCATCATCC
IL-6	F: TCACAGAAGGAGTGGCTAAGG
	R: GCTTAGGCATAGCACACTAGG
TNF-α	F: CATCTTCTCAAAACTCGAGTGACAA
	R: TGGGAGTAGATAAGGTACAGCCC
STK4	F: TCCGAGTAGCCAGCACGATGAG
	R: GGTTCCTTCCTCTTCCTCGTCCTC
IRAK1	F: GCGTAGCTGACCTCGTTCACATC
	R: GGAGAGGAAGGTGGAGGCAGAG

## Data Availability

The datasets used and/or analyzed during the current study are available from the corresponding author upon reasonable request.

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
