# Peer review of "Saikosaponin-b2 Inhibits Primary Liver Cancer by Regulating the STK4/IRAK1/NF-κB Pathway"

_biomedicines, 2023, doi:10.3390/biomedicines11102859_

Round 1
Reviewer 1 Report
In this manuscript, the authors demonstrated the anti-cancer effects of saikosaponin-b2 (SS-b2) on primary liver cancer and anti-inflammatory function on macrophages. In addition, they found that SS-b2 treatment increased and decreased the expression of STK4 and IRAK1 in the livers and macrophages. The experiments are well designed and performed, and the findings are or interest. Moreover, the data sufficiently support the conclusion. However, I found several issues that need to be addressed.1. Methods how to isolate peripheral macrophages are not described. Please rationalize in the introduction or result sections why peripheral macrophages were used to determine IRAK1 and STK4 mRNAs.
2. The method section lacks detail information regarding antibodies (clonality, clone name, company...). NFkB is the name of heterodimer. Please specify the target of the NFkB antibody.
3. line 189: Tukey's test should be used for multiple comparison. Comparison between two groups should be performed by Student's, Welch's or Mann-Whitney's tests.
4. Figure 1EFG, 2AB: Resolution of those images is low.
5. Figure 2AB: Which cells showed increased STK4 and decreased IRAK1 expression?
6. Figure 4G: Pearson's or Spearman's analysis should be performed.
7. Liver sections of DEN-treated mice with or without SS-b2 administration should be stained with macrophage markers (F4/80...), and if possible, with M1/M2 marker (CD80 or MHC class II/CD206) to determine whether tumor-associated macrophages are involved in the anti-cancer effect of SS-b2.
8. This study demonstrated the preventive effect of SS-b2 on the development of primary liver cancer. However, most persons take medications after diagnosis of liver cancer. In that case, SS-b2's anti-inflammatory effects would possibly disturb its therapeutic effect on liver cancer. I would like the authors to discuss regarding its therapeutic potential against HCC and this concern.
Author Response
We greatly appreciate your insightful and helpful suggestions. We have revised our manuscript following your suggestions and instructions, after considering your feedback. Moreover, revised manuscript has also been reviewed by a native English speaker to make sure that no grammar or other language problem exits. Our point-by-point responses to the editor's comments are presented in following document.

Reviewer 2 Report
This manuscript described the preventative effect of Saikosaponin-b2 (SS-b2) on the development of primary liver cancer (PLC) in the diethylnitrosamine (DEN)-induced liver cancer model of mice. Furthermore, the authors claimed that SS-b2 retards PLC development by suppressing DEN-induced liver inflammation through upregulating STK4, which presumably binds to IRK4 to promote IRK4 degradation, leading to the inhibition of IRK4-mediated NF-kappaB activation and the resultant suppression of inflammation. Comments are provided as follows.
1. The main concern in this manuscript is that the presented evidence merely reveals the correlation between SS-b2’s anti-liver cancer effect with inhibition of IRK4/NF-kappaB through STK4 upregulation. There is no direct evidence to support the conclusion that SS-b2 prevents PLC by negatively regulating IRAK4/NF-kappaB signaling axis through upregulating STK4. To make this argument, the authors need to confirm this mechanism in the context of an in vitro liver cancer model. In other words, the authors need to prove that STK4 knockdown would upregulate IRAK4 and thus the activation of NF-kappaB activity revealed by the increase in the levels of proinflammatory cytokines, consequently impairing SS-b2-induced inhibition of liver cancer development.
2. Please avoid using the term “STK4/IRAK4/NF-kappaB signaling pathway” throughout the manuscript. It is better to replace it with a statement such as “SS-b2 upregulates STK4 to suppress the IRAK4/NF-kappaB signaling axis”.
3. The increase in STK4 and decrease in IRK4 shown in the Western blot images in either the in vivo liver tissues or in LPS-treated RAW 264.7 cells were very limited. The authors need to present more convincing evidence of Western blot images.
4. Please define the NF-kappaB examined in this study. Specifically, please clarify the subunit of NF-kappaB examined by the IHC and western blot analysis in this study.
5. The authors examined the mRNA levels of STK4 and IRAK4 in the peripheral blood macrophages. Then they jumped to the conclusion that “These results suggest that SS-b2 may prevent the malignant development of PLC by regulating the STK4/IRAK1 pathway, thereby inhibiting downstream NF-κB activation and inflammatory factor production” (Line 275-276). How can the inverse relationship between STK4 and IRAK4 in the peripheral blood macrophages cause the SS-b2’s inhibitory effect on tumor formation in the liver?
6. Please define the “degree of malignancy” and “liver index” shown in the manuscript. In particular, what are the parameters to evaluate malignancy? Can it be expressed in a quantitative way, such as a scoring system?
7. Please expand the background information on IRK4 in the Introduction section. In particular, how IRK4 regulates NF-kappB activity and its role in promoting inflammation. The clinical implication of IRK4 in liver cancer development should also be justified.
8. Line 12: please replace “antioncogenes” with “tumor suppressor genes”.
9. Line 14: please replace “regulation” with “effect”.
English editing is recommended.
Author Response
We greatly appreciate your insightful and helpful suggestions. We have revised our manuscript following your suggestions and instructions, after considering your feedback. Additional experiments were performed and results from these experiments have been added as suggested (See Fig. 4, Fig. 5E). The data summarized in our manuscript showed that STK4 knockdown would upregulate IRAK4 and thus the activation of NF-kappaB activity revealed by the increase in the levels of proinflammatory cytokines, consequently impairing SS-b2-induced inhibition of liver cancer development. Additionally, SSb2 significantly inhibited HepG2 liver cancer cell proliferation. These results suggest that SS-b2 effectively inhibited PLC by upregulating STK4 to suppress the IRAK4/NF-κB signaling axis. Moreover, revised manuscript has also been reviewed by a native English speaker to make sure that no grammar or other language problem exits. Our point-by-point responses to the editor's comments are presented below.

Round 2
Reviewer 1 Report
1. The method for peripheral macrophage isolation is not appropriate for this purpose from some reasons. 1) Almost no macrophages are present in peripheral blood. Macrophage require to be differentiated from peripheral blood monocytes.
2) So far as I know, this kind of leukocyte separation solution is based on specific gravity separation; thus, it should not be mixed thoroughly, but should be overlaid.
Therefore, taking this into consideration, the methods and results should be revised.
2. Methods of statistical analyses should be shown every figure legend.
Author Response
Thank you for your thoughtful comments and suggestions. We have revised our manuscript based on your suggestions and instructions and carefully considered your comments. Our point-by-point responses to the reviewer's comments are presented below. Please see the attachment.

Reviewer 2 Report
The Reviewer thanks the authors' efforts to address my concerns on the original manuscript. Most of my concerns have been addressed satisfactorily. Still, the following comments should be addressed before the acceptance of this manuscript:
1. The authors demonstrated STK4 knockdown led to increased IRAK1 and p65 using HepG2 as liver cancer a cell model. However, the STK4 levels were not increased in the SS-b2-siRNA-NC experimental group shown in the western blot result in Figure 4C, which is inconsistent with the results shown in Figure 4D. Please justify this apparent difference.
2. The authors must prepare Figure 4C more carefully; particularly, the labels must be correct. For example, the label of the 2nd lane in Figure C should be siRNA-STK4. Likewise, the 2nd label in Figure 4D should be siRNA-STK4.
3. Likewise, the RAK1 shown in the legend of Figure 2 should be corrected as IRAK1.
4. The authors stated in the Introduction that STK4 downregulates IRAK1 by promoting IRAK1 protein degradation, leading to the downregulation NF-kappB activation. However, the authors examined the mRNA expression levels of IRAK1 and NF-kappB instead of IRAK1 protein stability and the phosphorylation levels of p65, which reveals NF-kappaB activation status. Please justify the rationale of the research approach.
5. The rationale for examining the STK4, IRAK4, and p65 levels in SS-b2 macrophages should be justified in the context of the role of macrophages in liver cancer development. Should it be about the effect of the tumor microenvironment on liver cancer development?
The description from Lines 307 to 308 stated "......the expression of STK4 decreased......" should be corrected as "......the expression of STK4 was decreased......".
Author Response

(The authors gave the same response as above.)

Round 3
Reviewer 2 Report
No more comments.
Minor English editing is recommended.